# Biochemical and Molecular Pathways in Neurodegenerative Diseases: An Integrated View

**DOI:** 10.3390/cells12182318

**Published:** 2023-09-20

**Authors:** Nitesh Sanghai, Geoffrey K. Tranmer

**Affiliations:** 1College of Pharmacy, Rady Faculty of Health Science, University of Manitoba, Winnipeg, MB R3E 0T5, Canada; geoffrey.tranmer@umanitoba.ca; 2Department of Chemistry, Faculty of Science, University of Manitoba, Winnipeg, MB R3T 2N2, Canada

**Keywords:** neurodegenerative diseases, Alzheimer’s disease, Parkinson’s disease, amyotrophic lateral sclerosis, ageing, oxidative stress, excitotoxicity, calcium butterfly effect, proteostasis, chaperones, autophagy, mitophagy, neuroinflammation

## Abstract

Neurodegenerative diseases (NDDs) like Alzheimer’s disease (AD), Parkinson’s disease (PD), and amyotrophic lateral sclerosis (ALS) are defined by a myriad of complex aetiologies. Understanding the common biochemical molecular pathologies among NDDs gives an opportunity to decipher the overlapping and numerous cross-talk mechanisms of neurodegeneration. Numerous interrelated pathways lead to the progression of neurodegeneration. We present evidence from the past pieces of literature for the most usual global convergent hallmarks like ageing, oxidative stress, excitotoxicity-induced calcium butterfly effect, defective proteostasis including chaperones, autophagy, mitophagy, and proteosome networks, and neuroinflammation. Herein, we applied a holistic approach to identify and represent the shared mechanism across NDDs. Further, we believe that this approach could be helpful in identifying key modulators across NDDs, with a particular focus on AD, PD, and ALS. Moreover, these concepts could be applied to the development and diagnosis of novel strategies for diverse NDDs.

## 1. Introduction

Neurodegeneration causes progressive loss of non-regenerative neurons in the brain and spinal cord [1]. Various biochemical pathways are implicated in the progression of neurodegenerative diseases (NDDs) like Alzheimer’s disease (AD), Parkinson’s disease (PD), and amyotrophic lateral sclerosis (ALS) [2,3]. Several complex pathologies have been reported over the years with converging biochemical cascades in neuronal death [4]. The most important biochemical perils include firstly the perturbed redox pathway in neuronal cells due to hypermetallation [5,6,7,8], leading to redox dyshomeostasis (Figure 1) [9,10,11]. Secondly, the butterfly effect of calcium-related dysfunctions in neurons leads to the propagation of excitotoxicity and hence, incites multiple pathologies in the progression of neurodegeneration (Figure 2) [12,13,14,15]. Thirdly, the defective protein quality control pathways in neurodegeneration lead to the aggregation of misfolded proteins in NDDs (Figure 3) [16,17,18,19,20]. Lastly, recent evidence from various scientific communities suggests that neuroinflammation plays a crucial role in the onset and progression of several NDDs [21,22,23].

Above all, ageing [8,24,25,26], which is a physiological process of human life, is one of the determinants of neuronal vulnerability and in many cases leads to the probability of increasing NDDs. Herein, we will explore the various common or overlapping biochemical molecular mechanisms and indispensable cross-talks implicated in NDDs.

## 2. Biochemical Pathways Perturbed by Different Metal Ions and the Pathological Role of Free Radicals in NDDs

The human brain weighs merely ~1400 g; however, it consumes ~20% of the total basal oxygen (O_2_) budget to power its ~86 billion neurons and their highly complex synapses, fueled by adenosine triphosphate (ATP) formed in mitochondria. The O_2_ we breathe is a mutagenic gas, due to its diradical and triplet spin state, and is implicated in the formation of the precursors of all free radicals via superoxide anion radical (O_2_**^•−^**) [27]. The (O_2_**^•−^**) undergoes a chemical redox reaction to produce reactive oxygen species (ROS) or reactive nitrogen species (RNS) including non-radicals, free radicals, and anions, such as hydrogen peroxide (H_2_O_2_), hydroxyl radical (HO^•^), and peroxynitrite (ONOO^-^), causing an imbalance in cellular homeostasis called oxidative stress (OS) (Figure 1). OS is largely implicated in the biochemical pathophysiology of NDDs like AD, PD, and ALS [9]. The most important factors leading to oxidative damage in neurodegeneration are inevitable ageing, the presence of redox transition metals, and excitotoxicity. Redox-active transition metals (RATM) in their reduced state (i.e., ferrous ion (Fe^2+^) and cuprous ion (Cu^+^) are enriched in the brain, and they take part in various chemical reactions, mainly with oxygen (O) and nitrogen (N) group of bio-molecules) [28]. Fe^2+^ is an essential RATM for myelin synthesis [29,30] and acts as a co-factor for essential *de novo* lipid synthesis enzymes [8]. Fe^2+^ regulates ferroptosis—a novel iron-dependent, non-apoptotic, and non-necrotic form of cell death due to (OS) caused by lipid peroxidation pathway, and is regarded as one of the potential causes for pathogenesis of various NDDs [31,32,33]. Like Fe^2+^, neurons contain a “labile” Cu^+^ pool [34]. Cu^+^ helps in neuronal excitability, due to the re-distribution of Cu^+^ from soma to dendrites. In addition, Cu^+^ is an essential co-factor for many enzymes like mitochondrial cytochrome c oxidase (CcO) [35], which acts as an electron acceptor in the electron transport chain (ETC), thus producing energy, ATP [36]. Cu^+^ has a catalytic role in the function of the ubiquitous antioxidant copper-zinc superoxide dismutase 1 (CuZnSOD, or SOD1) enzyme. Further, the Cu^+^ chaperone for SOD1 (CCS), is crucial for Cu^+^ insertion and disulfide (-S-S-) bond formation [37]. CCS prevents the accumulation of misfolded mutant SOD1 and promotes zinc (Zn) binding, which has a structural role to play in the SOD1 function [38]. Recently, evidence has shown that Cu^+^ could block glutamate receptors [39,40].

Ageing is inevitable and is regarded as one of the primary risk factors for the degeneration of post-mitotic neuronal cells in the CNS [25,41]. One of the most important causes of selective neuronal vulnerability (SNV) during ageing is due to the increase in metals, like Cu^+^ and Fe^2+^. These RATM are known to cause neuronal death due to an increase in ROS and hypermetallation in misfolded toxic aggregates of proteins (Aβ in AD and α-synuclein in PD) during the process of ageing [42,43,44]. However, SOD1 aggregation in the case of familial ALS (fALS) is due to aberrant post-translational modifications (PTMs) [45] instigated by demetallation leading to loss of Cu^+^, whereas Fe^2+^ was shown to be increased in ALS pathology [46]. These abnormal toxic proteins (Aβ) in AD [44], α-synuclein in PD [47,48], and SOD1 in ALS [46], abnormally present Fe^2+^ and Cu^+^ ligands for inappropriate chemical reactions with H_2_O_2_ called Fenton and Haber–Weiss reactions, respectively (Figure 1) [9]. Both these chemical redox reactions produce nature’s most vulnerable hydroxyl radical (HO^•^), which accelerates the process of misfolding and hence, the formation of toxic aggregates leading to neuronal death. Several studies over the past three decades have decoded the toxic role of H_2_O_2_ in the pathogenesis of NDDs [49]. H_2_O_2_ can display both Jekyll and Hyde behavior as a stable ‘diffusible’ non-ionized oxidant in living cells. It acts as a double-edged sword molecule, depending upon the physiological concentration. Lower concentration, called physiological concentration in the range of (1–10 nM), acts as a redox cell signaling molecule in various biochemical cellular processes, creating oxidative eustress. Higher (or pathological) concentration of H_2_O_2_ of around (>100 nM) is known to cause damaging effects on cellular biomolecules; this effect is called oxidative distress and acts as a bio-precursor for generating toxic oxidant (HO^•^) radicals. These radicals can act as a determinant to trigger the biochemical conformational trajectories via changing the cellular redox thiol (SH) status of several proteins leading to misfolding and toxic proteinopathies in NDDs like amyloid β in AD [50,51], α-synuclein in PD [52,53] and SOD1 [54,55,56], and TDP-43 in case of ALS [54,57,58].

Mounting evidence has shown the presence of OS biomarkers [59,60,61] in NDDs [62,63,64,65] due to the damaging effect of (HO^•^) radicals. Elevated 4-hydroxynonenal (HNE) levels have been observed in AD [66] and PD [67] brain tissue, whereas increased HNE has been observed in the cerebrospinal fluid (CSF) of ALS patients [68]. Thiobarbituric acid-reactive substances (TBARs) have been observed in AD [69], PD [70], and ALS [71]. The oxidative lipids acrolein and HNE induce toxicity by crosslinking to cystine, lysine, and histidine residues via a Michael addition [9]. Recent evidence has shown the generation of (HO^•^) by a Fenton-like reaction involving Fe^+2^ with histidine complex in the case of AD [72]. Further, oxidation of selected histidine residues, such as 2-oxohistidine, binds metals in the active site and can mediate SOD1 aggregation in ALS [73]. In the case of PD, the dopaminergic neurons containing dopamine neurotransmitters undergo Fenton-like reactions to produce oxidative metabolites, like dopamine quinones and (HO^•^), and cause neurotoxicity [74]. 8-hydroxyguanosine (8-OHG) and 8-hydroxy-2-deoxyguanosine (8-OHdG) are observed as biomarkers for nucleic acid deoxyribonucleic acid/ribonucleic acid (DNA/RNA) oxidation in the brains of AD [75], PD [76], and ALS [77] patients. Further, protein carbonylation as a result of protein oxidation is found in AD [78], PD [79], and ALS [80], (Figure 1). On the other hand, the brain uses neuronal nitric oxide synthase (nNOS) and nicotinamide adenine dinucleotide phosphate NAD(P)H oxidase (NOX) biochemistry, forming peroxynitrite anion (ONOO^−^) for cell signaling. However, during redox dyshomeostasis of neurological disorders like AD [81], PD [82], and ALS [83], high levels of ONOO^−^ forms 3-nitrotyrosine (3-NO2Tyr), which act as a versatile biomarker of nitrosative stress (Figure 1) [84,85].

Another metal, Zinc ion (Zn^+2^), is a well-known redox-inert metal and helps in neurogenesis, neuromodulation, and axonal and synaptic transmission [86]. Zn^+2^ is found in the ubiquitous antioxidant enzyme SOD1, where it maintains the structural integrity of the enzyme and inhibits Fenton’s chemistry via inhibition of nicotinamide adenine dinucleotide phosphate oxidase (NADPH-Oxidase) [87,88]. The expression levels of the Zn^+2^ transporter are altered abnormally during AD; more importantly, ZnT3 levels were further decreased in the AD cortex. Thus, synaptic zinc release may be decreased in AD [89]. Further, Zn deficiency leads to the accumulation of α-synuclein, leading to toxicity in PD [90,91]. Recent studies have shown that Zn loss due to demetallation during abnormal PTMs leads to misfolding and gain of toxic function in the pathology of ALS [57,92,93,94].

These reports from the various scientific literature over the years have shown the critical importance of redox balance in the CNS. Further, various metal ions and bio-reactive free radicals mentioned above act as a common determinant in changing the redox signaling, thus initiating pathological processes in the degeneration of neurons in the case of AD, PD, and ALS. Moreover, these metals and OS biomarkers serve as a common hallmark of neurodegeneration across all NDDs.

**Figure 1 cells-12-02318-f001:**
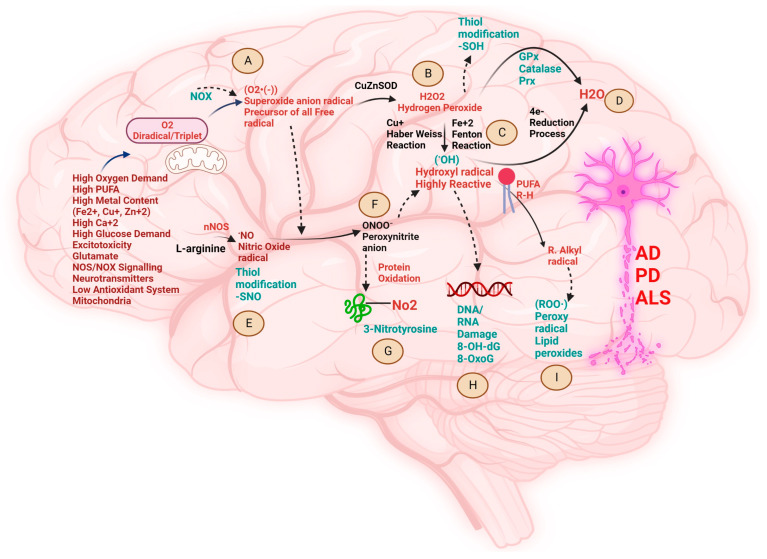
Schematic presentation of various biochemical cross-talks and their detrimental manifestations (**A**–**I**) in the brain provoked by oxidative stress and their implications in the progress of neurodegenerative diseases like Alzheimer’s disease (AD), Parkinson’s disease (PD), and amyotrophic lateral sclerosis (ALS). Brain is highly vulnerable to oxidative stress due to low regenerative capacity, enrichment of polyunsaturated fatty acids, high dependency on mitochondria for adenosine triphosphate (ATP) generation, elevated glucose demand, high concentration of metals like ferrous ion (Fe^+2^), cuprous ion (Cu^+^), zinc ion (Zn^+2^) and calcium ion (Ca^+2^), glutamate-induced excitotoxicity, high oxygen (O_2_) consumption, and relatively low antioxidant system. These multiple factors initiate various reaction pathways to create redox disbalance called oxidative and nitrosative stress in the brain, implicated in various NDDs. (**A**)**.** The triplet unstable O_2_ undergoes reduction to produce the precursor of all radicals called superoxide anion radical (O_2_^•−^) via NAD(P)H oxidases (NOXs) pathway, i.e., one-electron trans-membrane transfer to (O_2_) [95]. (**B**)**.** Antioxidant superoxide dismutase (SOD1) undergoes dismutation to scavenge (O_2_^•−^) to produce hydrogen peroxide (H_2_O_2_). (**C**)**.** The weakly liganded (Fe^+2^) and (Cu^+^) undergo reduction to produce nature’s most vulnerable oxidant hydroxyl radical (HO^•^) through Fenton’s reaction and Haber-Weiss reaction. (**D**)**.** The final 4th electron reduction of H_2_O_2_ in the presence of antioxidants, like glutathione peroxidase (Gpx), catalase (cat), and peroxiredoxin system (Prx), forms water (H_2_O). (**E**)**.** Overactivation of neuronal nitric oxide synthase (nNOS) produces nitric oxide (NO^•^) radicals from L-arginine, which create nitrosative stress by modification of thiol group (SH) containing proteins. (**F**)**.** Excessive superoxide anion radicals lead to inactivation of nitric oxide production and switch the biology to production of highly potent oxidant peroxynitrite anion (ONOO^−^), which leads to the nitrosative stress by (SH) modification of free tyrosine (Tyr) residues to form 3-nitrotyrosine (3-NO2Tyr) (**G**), which act as a versatile biomarker of nitrosative stress and NDDs. (**H**)**.** Highly reactive and mutagenic oxidant (HO^•^) damages the nucleic acid deoxyribonucleic acid/ribonucleic acid (DNA/RNA) to form oxidative products 8-hydroxy-2′-deoxyguanosine(8-OHdG) and (8-OxoG), and acts as a universal biomarker for oxidative stress and NDDs (important to note that guanine is the most oxidation prone nucleobase because of low reduction potential [96]). Further, HO^•^ radical causes lipid peroxidation of lipid-rich neuronal membranes, resulting in the death of neurons. Lipid peroxides (ROO**^.^**) act as a biomarker of oxidative stress and NDDs. Created with BioRender.com (accessed on 19 September 2023).

## 3. Biochemical Pathways and Cross-Talk between Excitotoxicity, Calcium ion (Ca^+2^), Fe^+2^, and Zn^+2^ in NDDs

The glutamatergic system is essential for brain functioning, with 40% of glutamatergic synapses located in the central nervous system [97,98]. Seminal work by John Olney provided the first evidence of the neurotoxic properties of the excitatory neurotransmitter glutamate. Since then, glutamate-driven neuronal death has been linked to several NDDs, like AD [99], PD [100], and ALS [101]. The biochemical mechanism that incites excitotoxicity involves alterations of glutamate receptors, mainly *N*-methyl-D-aspartic acid receptors (NMDAR), highly permeable to (Ca^+2^) and sodium ion (Na^+^) [12]. The exacerbated or prolonged activation of glutamate receptors starts a cascade of biochemical molecular pathways, which includes cationic influx, mitochondrial dysfunction, oxidative stress, and overproduction of ROS [102]. Mounting evidence has shown the role of calcium ions (Ca^2+^) to be critical in the biochemical pathways of NDDs, involving excitotoxic neurotransmitter glutamate, Zn^+2^ [103] and Fe^+2^ [104], and the cascade called neurotoxic excitotoxicity cascade (Figure 2) [103,105]. Glutamate homeostasis in the synaptic cleft is maintained by astrocytes and further, they are involved in 90% of glutamate clearance; during an acute insult, astrocytes can impede excitotoxicity by eliminating extracellular glutamate with high-affinity sodium-dependent glutamate transporters, also known as excitatory amino acid transporters (EAAT) [106,107,108]. Glutamate-induced excitotoxicity may be encouraged through an astrocyte-mediated downregulation of excitatory amino acid transporter 2 (EAAT2). In addition, astrocytes can also modulate the susceptibility of motor neurons to excitotoxic insults by regulating the influx of calcium through alpha-amino-3-hydroxy-5-methylisoxazole-4-propionic acid (AMPA) receptors [109]. Astrocytes act as gatekeepers for the maintenance of glutamate homeostasis by supporting its biosynthesis, uptake, and release via the glutamate-glutamine cycle [110,111]. Furthermore, astrocytes are responsible for the synthesis of lactate, which is taken by neurons for energy production via the citric acid cycle because of the absence of the essential enzyme pyruvate carboxylase [112,113,114]. Moreover, astrocytes are the main source of D-serine, essential for NMDAR function [115]. Therefore, growing evidence has demonstrated the biochemical role of excitotoxicity induced by astrocytic dysregulation in AD [99,116,117], PD [100,118,119], and ALS [109,120,121] (Figure 2). Further, the vicious cycle induced by glutamate-induced excitotoxicity and its disruption of (Ca^+2^) homeostasis thus accelerates oxidative and nitrosative stress in mitochondria and endoplasmic reticulum (ER) and hence, forms a quartet to initiate degeneration of neurons in NDDs [122,123]. This quartet leads to proteinopathies like SOD1 [124,125,126] and TDP-43 [127,128] in the case of ALS, β-amyloid protein in the case of AD [129,130,131,132], and α-synuclein in the case of PD [133,134].

The above-mentioned scientific evidence from various works of literature has concluded the cross-link between excitotoxicity, Ca^+2^, Fe^+2^, and Zn^+2^ across NDDs. The increased glutamate acts as a first pathological signaling in instigating a myriad of overlapped pathological cascades in various NDDs. Further, the excitotoxic degeneration of neurons is due to the Ca^+2^, which acts as an indisputable signaling metal exerting the butterfly effect. Hence, it serves as an impetus for initiating a cascade of pathological neurodegenerative processes in association with Fe^+2^ and Zn^+2^ in case of AD, PD, and ALS. In addition, these biochemical signatures could act as a therapeutic target in halting the progression of NDDs.

**Figure 2 cells-12-02318-f002:**
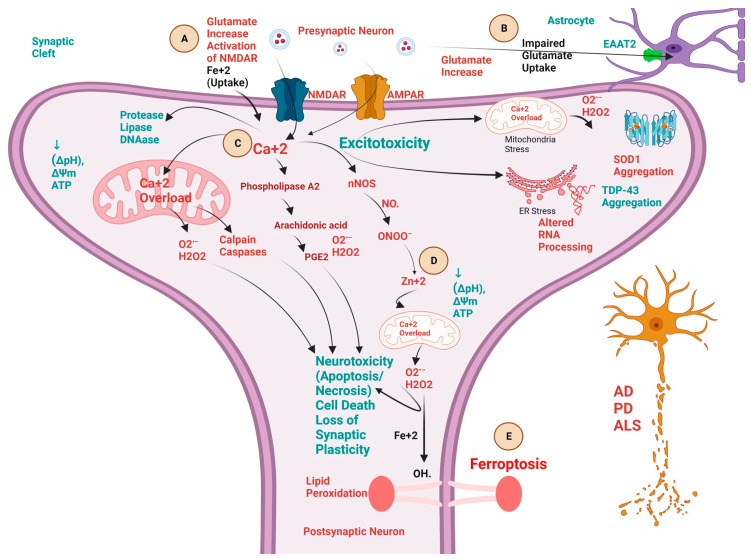
Schematic presentation of various biochemical cross-talks, involving calcium ion (Ca^+2^), ferrous ion (Fe^+2^), and Zinc ion (Zn^+2^) implicated in the progress of neurodegenerative diseases like Alzheimer’s disease (AD), Parkinson’s disease (PD), and amyotrophic lateral sclerosis (ALS). (**A**). Excitotoxicity (neuronal death) is triggered by the excessive release of excitatory neurotransmitter glutamate (neurotoxic) from the presynaptic neuron and leads to activation of various biochemical cascades leading to neurotoxicity and hence, neuronal death. This process is initiated by the activation of the *N*-methyl-D-aspartic acid receptors (NMDAR) by excessive glutamate at postsynaptic neurons and thereby the release and accumulation of toxic intraneuronal Ca^2+^. (**B**). Glutamate-mediated excitotoxicity is increased because of the astrocyte-mediated downregulation of excitatory amino acid transporters 2 (EAAT2), which slows down the uptake of glutamate from the synaptic cleft and incites the excitotoxicity cascade. (**C**). Ca^2+^ overload initiates most of the deleterious downstream mechanisms of the cascade, through increasing Ca^2+^ overload in mitochondria, induction of proteases (calpains and caspases), decreasing the proton gradient (ΔpH), mitochondrial membrane potential (ΔΨm) and adenosine triphosphate (ATP), activation of phospholipase A2 (PLA2) pathway initiating downstream activation of arachidonic acid and prostaglandin E2 (PGE2), aggravation of mitochondrial and endoplasmic reticulum stress leading to superoxide dismutase (SOD1) and TAR DNA-binding protein (TDP-43) aggregation. (**D**). Surge of reactive oxygen species (ROS) like hydrogen peroxide (H_2_O_2_) and hydroxyl radical (HO^•^) and reactive nitrogen species (RNS) like nitric oxide (NO^•^) radical, formation of peroxynitrite anion (ONOO^−^) increases the intraneuronal Zn^2+^ mobilization, which targets mitochondria and further exacerbates Ca^2+^ dysregulation and ROS production. (**E**). Ca^+2^ and Fe^+2^ dysregulation participates in the ferroptosis death of neurons. Iron dysregulation leads to Ca^2+^ dysregulation and vice versa. Excessive glutamate increases the Fe^+2^ intake inside the neurons, thereby leading to excitotoxicity and lipid peroxidation via Fenton’s reaction called Ferroptosis. Created with BioRender.com.

## 4. Biochemical Pathways Involving Protein Homeostasis, Autophagy, Mitochondrial Homeostasis, Axonal Transport, Protein Seeding, and Propagation and Their Implication in the Pathophysiology of NDDs

Recent evidence from a large number of groups has shown that there are common cellular and pathological mechanisms among numerous NDDs, which include converging biochemical mechanisms, such as defective protein quality-control and degradation pathways, dysfunctional mitochondrial homeostasis, stress granules, and abnormal innate immune responses. Despite their common biochemical pathways, they show loss of specific neurons and synapses in distinct brain regions [4]. One of the most common hallmarks is the aggregation of cytosolic and nuclear proteins due to dysfunction in protein homeostasis (called proteostasis), which causes neurodegeneration [16,135,136,137,138]. During the course of these proteinopathies, beta-amyloid (Aβ) aggregates in AD, inclusions of hyperphosphorylated microtubule-binding tau in AD and other tauopathies, aggregates of α-synuclein in PD and other synucleinopathies, and inclusions of TAR DNA-binding protein (TDP)-43 in case of ALS occur. Moreover, some toxic aggregates seed and spread from one region to another, consistent with the progressive nature of NDDs [4,139]. However, not all aspects of NDDs are the same because of the uniqueness in the genetic mutations in gene loci. Unlike other cells, neurons are post-mitotic cells, and they cannot divide and face several challenges in terms of continuous demand for energy production, maintenance of protein and organelle quality control, rapid delivery of molecules within and out of cells, and trafficking of organelles and other factors over considerable distances within the cell. Compromised pathways responsible for these functions can lead to NDDs.

Proteins must fold into well-defined 3D structures and need to remain folded and undergo quality control throughout their lifetimes to perform their biological functions. The state of balanced proteome homeostasis is called proteostasis and is governed by an extensive network of molecular chaperones, proteolytic systems, and their regulators, comprising ~2000 proteins in human cells [138]. One of the most essential parts of proteostasis is the presence of chaperones, which, with the help of ATPs, maintains proper protein folding and conformational maintenance without being part of its final structure and cooperate with the degradation machinery [16]. They are classified into small heat shock proteins (sHSPs). In mammals, Hsp90 helps in folding and conformational regulations. The Hsp70 major chaperone family is required for aggregation prevention, folding, and conformational maintenance, and it also cooperates with Hsp40 in the protein disaggregation or protein turnover of NDD proteins through the ubiquitin-proteasome system (UPS) (Figure 3) [4]. Autophagy (macroautophagy) is a catabolic process and is the cellular way of cleaning out damaged cells to regenerate newer and healthier cells. It is one of the most pivotal systems, and without it, the nervous system cannot function well [140]. Also, the activation of this self-destruction pathway is controlled by complex signaling mechanisms, which could be globally classified as mammalian targets of rapamycin (mTOR)-dependent or -independent pathways. mTOR modulates autophagy by suppressing the autophagic induction pathways [141], mainly via modulating the ULK1 ubiquitylation [142]. Autophagy is the regulator of misfolded aggregate-prone defective and toxic proteins that cause NDDs; for instance, mutant α-synuclein in PD [143], mutant TDP-43 in ALS [144], and Aβ in AD [145]. Abnormal degradation pathways due to defective autophagy could also lead to cell-to-cell propagation of toxic aggregates in the adjacent neurons in the central nervous system (CNS), causing the progression of AD [145,146], PD [147,148,149], and ALS [150,151,152] (Figure 3). The clearance of such substrates is retarded when autophagy is compromised, and clearance is induced when autophagy is stimulated. An autophagic receptor/adaptor like p62 is involved in the aggregation of Aβ, tau in the case of AD [153], α-synuclein in the case of PD [154], TDP-43 [155], and SOD1 in the case of ALS [156]; whereas optineurin (OPTN) is involved in the aggregation of tau in the case of AD and SOD1 and TDP-43 in the case of ALS. The autophagy gene *BECN1*, encoding the mammalian orthologue of the yeast *Atg6* (Beclin-1), has reduced messenger RNA (mRNA) levels in AD brain tissue [157]. Further, mutations in the ALS-causing gene *DCTN1* lead to impaired dynein/dynactin motor protein function, causing defects in the transport of autophagosomes, inducing axonopathy [158], and hence, motor neuron degeneration [159,160,161]. Moreover, the depletion of dynein/dynactin motor protein leads to neuromuscular synapse instability and functional abnormalities in both sporadic (sALS) and fALS [160,162,163] (Figure 3).

A significant topic implicated in the pathogenesis of NDDs is defective mitophagy [164,165,166,167,168]. It is defined as the selective autophagy or turnover of mitochondria. It is important for cells to maintain mitochondrial quality control through mitophagy and mitochondrial dynamics (fission and fusion). A progressive reduction in Parkin expression was observed in both AD patient brains as well as mutant human amyloid precursor protein transgenic mice (hAPP Tg) mouse models, suggesting an impairment in the effective activation of Parkin-mediated mitophagy during disease progression [79]. Further, levels of mitophagy-related proteins such as BCL2-like 13 (apoptosis facilitator) (Bcl2L13) and PTEN-induced kinase 1 (PINK1) downregulated in the hippocampal area of AD patient brains, in induced pluripotent stem cells (iPSC) derived cortical neuronal cultures generated from AD patients, indicative of a defective mitophagy pathway [169]. In the case of PD, mutations in PINK1 and Parkin, which are the predominant proteins involved in mitophagy, were shown to contribute to the early onset of autosomal recessive PD [170,171]. Moreover, in the case of ALS levels of mitophagy, proteins like Parkin, PINK1, Bcl-2 interacting protein 3 (BNIP3), and p62 were also found to be reduced in SOD1G93A mice [172]. In addition, studies in TDP-43^Q331K^ transgenic mice revealed dysregulations of Parkin and PINK1 mitophagic pathways in TDP-43 proteinopathy [173].

Together, these findings highlight the impairment of the mitophagy pathway contributing to the pathophysiology of NDDs (Figure 3). Further, rising evidence has shown the dysfunction of axonal transport (anterograde and retrograde) in the case of NDDs, such as in AD [174,175,176,177], PD [177,178,179], and ALS [180,181,182,183] (Figure 3). The cytoskeleton of large projection neurons might be particularly prone to dysfunction, as suggested by the biochemical pathways of aggregation and displacement of axonal neurofilaments (Nf) proteins and the microtubule-associated protein tau, observed in motor neurons in ALS and pyramidal neurons in AD [184]. Moreover, axonopathy in cases of neurodegeneration, like AD [185,186], PD [187,188], and ALS [189,190], causes the release of phosphorylated neurofilament (pNf) into the cerebrospinal fluid (CSF), and subsequently into the blood. Thus, increased neurofilament light chain (NfL) in biofluids acts as a potential biomarker in NDDs like AD, PD, and ALS. Specifically, in the case of AD levels of NfL [186,191,192], NfL during the progression of PD [193,194], and in the case of ALS, NfL and phosphorylated neurofilament heavy chain (pNFH) [190,195,196] reflect global neuronal axonal injury and, therefore, act as a prognostic biomarker for diagnosis of AD, PD, and ALS (Figure 3).

As described here, abnormal proteostasis, including (chaperones, autophagy, mitophagy, and proteosome) networks, acts as a decisive characteristic feature of neuronal death in the case of AD, PD, and ALS. Further, the increase in the propensity of progression of these NDDs is directly proportional to the defective PQC, leading to neurotoxic aggregation of proteins in NDDs. Moreover, specific genome instability leads to proteinopathies in a concerted fashion across various neurodegenerations. Thus, understanding and driving deep into the protein homeostasis pathways could give us an understanding of deciphering the therapeutic target across NDDs. The new approach could be identifying the defective protein signatures across various NDDs by utilizing global proteomics technique.

**Figure 3 cells-12-02318-f003:**
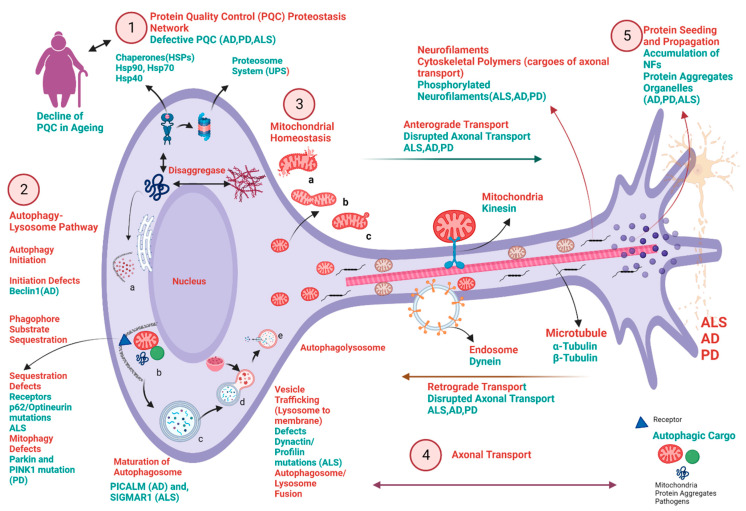
Schematic presentation of various common neuronal biochemical pathways perturbed or compromised in multiple neurodegenerative diseases, such as AD, PD, and ALS. The key points in the pathway and the selected disease-associated proteins are demonstrated in this picture. 1. Protein Quality Control (PQC) proteostasis network: molecular chaperones, including heat shock proteins (Hsp90, Hsp70, and Hsp40), regulate protein folding and maturation. Ubiquitin-proteosome system (UPS) is a crucial protein degradation pathway and is important for PQC and homeostasis. Any defect in the PQC leads to neurodegeneration (AD, PD, ALS). Decline of proteostasis is the hallmark of ageing and it decreases with age, leading to the accumulation of toxic and non-functional aggregates. 2. Autophagy-Lysosome Pathway (a,b,c,d,e.): Perturbations throughout the pathway, from initiation of autophagosome formation to degradation in the autolysosomes, have been suggested to be involved in neurodegenerative diseases like AD, PD, and ALS and further, could build an accumulation of pathogenic and toxic protein aggregates and defective mitochondria. a. Autophagy initiation defects due to decreased expression of protein Beclin1 in case of AD. b. Loss of sequestration into autophagosomes due to mutations in the gene-encoding p62/optineurin in the case of ALS, and mitophagy defects due to mutations in the gene-encoding protein PINK1/Parkin in the case of PD c. Defects in the maturation of autophagosome are due to decreasing expression of PICLAM protein in the case of AD, whereas mutation in SIGMAR1 gene in the case of ALS. c. Defects in vesicle trafficking (lysosome to membrane) are due to the mutations in the gene-encoding protein dynactin/profilin in case of ALS. 3. Dysregulation of mitochondrial quality control (MQC): including a (mitochondrial damage), b (mitochondrial fusion and fission dynamics), c (selective autophagy of mitochondria called mitophagy) results in decreased ATP production and dysfunctional proteostasis network. 4. Axonal transport defects in AD, PD, and ALS and underlying mechanisms: Defective axonal transport is due to perturbed anterograde and retrograde transport mechanisms involving mitochondrial kinesin and endosomal transport protein dynein. Further, disrupted neurofilament (NF) in forms of phosphorylated NF in the case of AD, PD, and ALS and microtubules (including α-Tubulin and β-Tubulin) are involved in the impairment of transport across neurons. 5. Protein Seeding and Propagation: Dysfunction of Intracellular propagation and seeding of toxic protein aggregates involved in the disease progression in case of AD, PD, and ALS. Created with BioRender.com.

## 5. Biochemical Pathways Altered in NDDs Due to Neuroinflammation

Neuroinflammation is a protective mechanism that initially protects the CNS from various pathogens and helps remove cellular waste and repair mechanisms [197]. However, extended periods of inflammation, which persist mainly due to changes in genetic makeup, neurotoxic protein aggregation, environmental pollution, infection, and exposure to drugs [198], could be detrimental and impair the regeneration of neuronal tissue in the CNS [199]. Microglia and astrocytes are the two main bio-inflammatory mediators associated with persistent neuroinflammation in the CNS [200]. Microglia are ubiquitously expressed immune cells in the CNS and are activated first in case of infections due to pathogens [201]. Microglia act as a mixed blessing depending upon the status of their stimulus. They could be pro-inflammatory or neuroprotective [202]. On the other hand, astrocytes are the indispensable glial cells in the brain and play an important role in maintaining CNS homeostasis [203]. Like that of microglia, astrocytes could be both pro-inflammatory and neuroprotective. The aggregation of neurotoxic proteins in NDDs, like amyloid-β (in case of AD) [204], tau (in case of AD) [204], α-synuclein (in case of PD), mSOD1 (in case of ALS), and TDP-43 (in case of ALS), initiates changes to induce both microglia and astrocytes to produce harmful pro-inflammatory pathological phenotypic biomarkers. The major pro-inflammatory biomarkers induced by both the glial cells implicated in the NDDs are interleukin-1 beta (IL-1β), tumour necrosis factor (TNF-α), interleukin 6 (IL-6), and nitric oxide (NO) [202,205]. Eventually, the production of these pro-inflammatory mediators leads to the progression of NDDs. Recent studies have shown that bivalent (Ca) plays an indispensable role in maintaining CNS homeostasis and aberrant Ca^+2^ signaling in the CNS leads to NDDs. This is largely evidenced by the abnormal Ca^+2^ dysregulation in the microglial and astrocytic cells of CNS, thus initiating a cascade of neuroinflammatory progression of the disease, like in AD [206,207], PD [208,209], and ALS [210,211].

Recent studies and perspectives have shown that nuclear factor κB (NFκB) in the case of AD [212], PD [213], and ALS [214] induces both microglia [215,216] and astrocytes [217,218] to produce several pro-inflammatory mediators implicated in various NDDs [200,219]. Further, free radical-generating enzymes such as cyclooxygenase-2 (COX-2), NADPH oxidase, inducible nitric oxide synthase (iNOS), and lipoxygenase are also implicated in NF-κβ activation in the case of AD [212]. Survival of motor neurons with the decrease in misfolded SOD1 protein has been reported recently, with neuronal inhibition of NF-κB activity in the SOD1G93A ALS mice model [220]. Together, enormous studies have portrayed the new modulatory role of NF-κB in instigating inflammation in the brain tissues [212,221,222,223,224,225].

To recapitulate, neuroinflammation contributes to both the onset and progression of disease in various NDDs. Glial cells and various other pro-inflammatory mediators mentioned above mediate neuroinflammation and thus neurodegeneration. With recent evidence, neuroinflammation in CNS is a common emerging factor in the case of AD, PD, and ALS. Further, studies are needed to understand and explore the biochemical factors that induce neuroinflammation. One such mediator of inflammation in neurons is NF-κB, which is now known to have profound effects in encouraging the neuroinflammation pathways. Further, suppression of neuroinflammation could ameliorate the symptom onset and progression of NDDs. One therapeutic target could be the modulation of neuroinflammation through modulating the deleterious effects of NF-κB in the CNS.

## 6. Conclusions

The brain is an organ that harbors post-mitotic neuronal cells which have no capacity to regenerate. Scientific evidence suggests that the brain is vulnerable to numerous insults due to ageing, oxidative stress, high metal content, poor quality control of organelles, dysfunctional proteostasis, excitotoxicity, defective axonal transport, and neuroinflammation. These complex cascades of biochemical pathways provoke the degeneration of neurons in the case of AD, PD, and ALS, together called NDDs. The pathophysiology of each NDD shares a common pathway with convergent cross-talks among various molecular mechanisms, because of common genotype-phenotype relationships among various NDDs.

A better understanding of these crucial biochemical pathological hallmarks across various NDDs like AD, PD, and ALS is crucial to finding therapeutic agents that could create hope in patients by slowing down the progression of NDDs. Various approaches could be used to slow down the progression of NDDs. Firstly, targeting a single etiological factor; however, due to multifactorial pathologies underlying NDDs, it would be difficult to alleviate the progression, targeting only one risk factor. Secondly, a comprehensive approach to target multiple etiological factors through a multitarget approach. We believe that targeting multiple aetiologies responsible for the disease could help slow and tackle the devastating and complex biochemical neurodegenerative cascade in the case of AD, PD, and ALS. Therefore, it is essential to understand various pivotal and common potential biochemical perils of cross-talks among various NDDs, which could be helpful in finding effective cocktail treatments for NDDs. Considering the complex and multifactorial nature of the NDDs, we further advocate designing novel molecules, with multi-targeted directed ligands having different pharmacophores, which could interact with different biomolecular targets or pathologies.

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
