# Peer review of "Biochemical and Molecular Pathways in Neurodegenerative Diseases: An Integrated View"

_cells, 2023, doi:10.3390/cells12182318_

Round 1

Reviewer 1 Report

Neurodegenerative disorders represent a group of progressive and debilitating conditions that primarily affect the central nervous system, leading to the gradual degeneration and loss of specific neuronal populations. These disorders, which include Alzheimer's disease, Parkinson's disease, Huntington's disease, and amyotrophic lateral sclerosis (ALS), pose significant challenges to healthcare systems worldwide due to their increasing prevalence with aging populations. Hence, understanding the underlying biochemical and molecular mechanisms that drive these diseases is crucial for the development of effective treatments and potential cures. Basic understanding of the aspects like, Protein Misfolding and Aggregation, Oxidative Stress, Mitochondrial Dysfunction, Excitotoxicity, Neuroinflammation, Autophagy Dysfunction, Genetic Factors, Prion-Like Propagation etc. might bring insight into novel therapeutics to treat Neurodegenerative disorders.

Researchers and clinicians are continuously working to unravel the intricate web of biochemical and molecular mechanisms involved in neurodegenerative disorders. Advancements in understanding these processes have opened up new avenues for developing targeted therapies, disease-modifying drugs, and potential biomarkers for early diagnosis and intervention. As the field progresses, a comprehensive understanding of the underlying mechanisms may lead to more effective treatments, potentially offering hope for those affected by these devastating conditions. Hence, the scientific aspect of the current review article is valuable and on demand. The authors here managed to summarize the key findings through text mining and referencing. Also, addition of attractive diagrams making the manuscript more informative and provide contextual reflection of biochemical and molecular aspect of Neurodegenerative disorders.

However, the manuscript lack a crucial aspect. Also certain titles are very casual, and need some modifications.  In the following points, I highlight parts of the manuscript I found to be less compelling, and I offer some suggestion.

1) The title of the review could be improved to make it more precise. While it is understandable, and it indicates it is a review, but could be a matter of personal taste.

Title Suggestion:

"Biochemical and Molecular Pathways in Neurodegenerative Diseases: An Integrated View"

Or

"Biochemical and Molecular Mechanisms Uniting Neurodegenerative Disorders: A Comprehensive Analysis"

2) The keywords should be more specific, and importantly the reflection of actual context, rather than general terminology. Based on texts the best keywords should be the following: Neurodegenerative diseases; Ageing; Oxidative stress; Excitotoxicity; Proteostasis

 3) The title (section 2) “Biochemical Pathways Perturbed by Oxidative Stress in NDDs” is very broad and not exactly reflecting the written context under the title. However, based on the text, title should be more precise. Authors here particularly focused on the biological role of different bivalent ions and pathological role of radical, hence authors must reflect the same through the title.

 4) Line 71-72, “There is a considerable probability and evidence of developing NDDs during the sixth, seventh and eighth decades of life.” Not very understandable, so need to reframe the sentence.

 5) The conclusion (section 4) is very casual. Need to add some prospective into it. Authors may add some aspects of diagnostics or therapeutics, using biochemical and molecular understanding of NDD with appropriate references. Also, In abstract authors mentioned (Line 18-20) “Moreover, these concepts could be applied to the development and diagnosis of novel strategies for diverse NDDs.” Authors might make another paragraph to summarize on this in the conclusion section, about the aspect on the how these understanding might be implemented in developing new therapeutic approach. That will make it more structured and appealing to read.

Author Response

Dear Reviewer,

My Sincere gratitude to you.

I want to thank and appreciate you, for reviewing our manuscript and making all the efforts to provide valuable comments and suggestions for the improvement of the manuscript. Further, we would like to thank you for making positive remarks on the manuscript, with interest in current topic and high quality figures. We also believe that this consolidated review will catch the attention of the wider scientific community.

Below are the comments (highlighted in bold) by the reviewer and the revisions (highlighted in red in the manuscript and below) according to the recommendations of the reviewers.

  • The title of the review could be improved to make it more precise. While it is understandable, and it indicates it is a review, but could be a matter of personal taste.

 Authors Comments: Thank you for the suggestions for the title. We have made changes to the title of the manuscript. Below is the revised title.

“Biochemical and Molecular Pathways in Neurodegenerative Diseases: An Integrated View”

  • The keywords should be more specific, and importantly the reflection of actual context, rather than general terminology. Based on texts the best keywords should be the following: neurodegenerative diseases; Aging; Oxidative stress; Excitotoxicity; Proteostasis

 Authors Comments: Thank you for the suggestions for the keywords. We have revised the keywords and added an additional keyword, neuroinflammation because we have added an additional paragraph on neuroinflammation according to the recommendation of reviewer two.

       “Neuroinflammation”

  • The title (section 2) “Biochemical Pathways Perturbed by Oxidative Stress in NDDs” is very broad and does not exactly reflect the written context under the title. However, based on the text, the title should be more precise. Authors here particularly focused on the biological role of different bivalent ions and the pathological role of radicals, hence authors must reflect the same through the title.

Authors Comments: Thank you for the suggestions. We have changed the subtitle. However, we cannot amend it to “bivalent ions” because copper is in cuprous form not in bivalent form. So, we have introduced the common term metals, considering the involvement of (Fe+2, Ca+2, Zn+2, and Cu+). Below is the revised sub-title.

Biochemical Pathways Perturbed by Different Metal Ions and the Pathological Role of Free Radicals in NDDs

  • Line 71-72, “There is a considerable probability and evidence of developing NDDs during the sixth, seventh, and eighth decades of life.” Not very understandable, so need to reframe the sentence.

Authors Comments: We have added the below sentence for better understanding, with citations.

Ageing is inevitable and is regarded as one of the primary risk factors for the degeneration of post-mitotic neuronal cells in the CNS [38, 39].

  • Moreover, these concepts could be applied to the development and diagnosis of novel strategies for diverse NDDs.” Authors might make another paragraph to summarize this in the conclusion section, about the aspect of how this understanding might be implemented in developing new a therapeutic approach. That will make it more structured and appealing to read.

Authors Comments: We have added a new paragraph about our approaches and perspective in the future for targeting NDDs. Below is the new paragraph added in the conclusion section according to the suggestions and comments.

The brain is an organ that harbours post-mitotic neuronal cells which have no capacity to regenerate. Scientific evidence suggests that the brain is vulnerable to numerous insults due to ageing, oxidative stress, high metal content, poor quality control of organelles, dysfunctional proteostasis, excitotoxicity, defective axonal transport, and neuroinflammation. These complex, cascades of biochemical pathways provoke degeneration of neurons in case of AD, PD, and ALS, together called NDDs. The pathophysiology of each NDDs shares a common pathway with convergent cross-talks among various molecular mechanisms, because of common genotype-phenotype relationships among various NDDs.

A better understanding of these crucial biochemical pathological hallmarks across various NDDs like AD, PD, and ALS is crucial to find therapeutic agents which could create hope in patients by slowing down the progression of NDDs. Various approaches could be used to slow down the progression of NDDs. Firstly, targeting a single etiological factor, however, due to multifactorial pathologies underlying NDDs, it would be difficult to alleviate the progression, targeting only one risk factor. Secondly, a comprehensive approach to target multiple etiological factors through a multitarget approach. We believe, that targeting multiple aetiologies responsible for the disease could be helpful in slowing and tackling the devastating and complex biochemical neurodegenerative cascade in the case of AD, PD, and ALS. Therefore, it is important to understand various pivotal and common potential biochemical perils of cross-talks among various NDDs, which could be helpful in finding effective cocktail treatments for NDDs. Considering the complex and multifactorial nature of the NDDs, we further, advocate designing novel molecules, with multi-targeted directed ligands having different pharmacophores, which could interact with different biomolecular targets or pathologies.

Reviewer 2 Report

Here, Authors describe the common biochemical and molecular mechanisms involved in neurodegenerative diseases, with the aim of identifying key modulators in disorders, such as Alzheimer’s disease, Parkinson’s disease, and amyotrophic lateral sclerosis, with the underpinning objective to characterize new targets for effective therapies. Authors deal with a topic of considerable interest, since, to date, few treatment options are available for neurodegenerative diseases. Even if the manuscript appears sound, it needs to be significantly improved to boost the interest of readers and give a contribution of novelty.

- A common pathogenic cause of great relevance to  neurodegeneration is neuroinflammation, which represents a common clue to a variety of neurodegenerative disorders, including Alzheimer's disease, Parkinson's disease, amyotrophic lateral sclerosis, and traumatic brain injury-induced neurodegeneration (doi: 10.1016/S1474-4422(15)70016-5, doi: 0.1186/s40035-020-00221-2, doi: 10.3390/cells11172728). This is completely ignored in the MS. The array of factors described in this review are tightly interconnected with overshooting inflammatory/immune response in the neurodegenerating brain, as well as in the periphery. To identify new therapeutic targets, it appears then crucial to explore the pathophysiological mechanisms generating neuroinflammation in neurodegenerative disorders. Therefore, Authors should add a paragraph about the biochemical, molecular and cellular mechanisms involved in neuroinflammation related to neurodegenerative diseases and on how altered inflammatory immune response may affect some (if not all) of the metabolic pathways object of the review. Doing so would be useful to highlight the novelty of this review over previously published summaries of potential common biochemical and molecular mechanisms involved in neurodegenerative diseases. Otherwise, the MS would sound as a obvious update of already widely considered mechanisms. The novelty should be endowed in linking the cited mechanisms to new vistas in the field, related to, for instance, proinflammatory/proapoptotic cytokines, immune cell trafficking through the blood brain barrier, etc, and how this would influence the equilibrium of different cells and molecules described in the MS.

- The quality of the figures is overall poor and thus needs to be significantly improved.

The manuscript needs to be better organized using subheadings and organizational paragraphs to lead the reader through the different topics. It should be revised for grammar and mistyping overall the text. Some examples:

- Line 9: “alzheimer’s” should be replaced with “Alzheimer’s”.

- Line 9: “parkinson’s” should be replaced with “Parkinson’s”.

- Line 167: “(Na)” should be replaced with “(Na+).

Author Response

Dear Reviewer,

My Sincere gratitude to you.

I want to thank and appreciate you, for reviewing our manuscript and making all the effort to provide valuable comments and suggestions for the improvement of the manuscript. We have added the paragraph for neuroinflammation and did all the efforts to demonstrate all the figures maintaining high standards using the widely accepted Bio Render tool.

Below are the comments (highlighted in bold) by the reviewer and the revisions (highlighted in red in the manuscript and below) here and in the manuscript according to the recommendations of the reviewers.

  1. A common pathogenic cause of great relevance to neurodegeneration is neuroinflammation, which represents a common clue to a variety of neurodegenerative disorders, including disease, Parkinson's disease, amyotrophic lateral sclerosis, and traumatic brain injury-induced neurodegeneration(doi: 10.1016/S1474-4422(15)70016-5, doi: 0.1186/s40035-020-00221-2, doi: 10.3390/cells11172728). This is completely ignored in the MS. The array of factors described in this review is tightly interconnected with overshooting inflammatory/immune response in the neurodegenerating brain, as well as in the periphery. To identify new therapeutic targets, it appears then crucial to explore the pathophysiological mechanisms generating neuroinflammation in neurodegenerative disorders. Therefore, the Authors should add a paragraph about the biochemical, molecular and cellular mechanisms involved in neuroinflammation related to neurodegenerative diseases and on how altered inflammatory immune response.

Authors Comments: Thank you for the suggestions to include the paragraph highlighting the important perspective on neuroinflammation and how it is linked to NDDs. We have added the below paragraph in the manuscript, highlighting important pro-inflammatory mediators and the new perspective on the role of Ca+2 and NF-κB in neuroinflammation. We have also, added the recommended citations in the paragraph, further, we have added twenty-eight citations for this additional paragraph to make it intriguing.

“Biochemical Pathways Altered in NDDs Due to Neuroinflammation”

Neuroinflammation is a protective mechanism that initially protects the CNS from various pathogens and is helpful in removing cellular waste and in repairing mechanisms [195]. However, extended periods of inflammation, which persist mainly due to changes in (genetic makeup, neurotoxic protein aggregation, environmental pollution, infection, and exposure to drugs)[196], could be detrimental and impair the regeneration of neuronal tissue in the CNS[197]. Microglia and astrocytes are the two main bio-inflammatory mediators associated with persistent neuroinflammation in the CNS [198]. Microglia are ubiquitously expressed immune cells in the CNS and are activated first in case of infections due to pathogens[199]. Microglia act as a mixed blessing depending upon the status of its stimulus. It could be pro-inflammatory or neuroprotective [200]. On the other hand, astrocytes are indispensable glial cells in the brain and play an important role in maintaining CNS homeostasis[201]. Alike that of microglia, astrocytes could be both pro-inflammatory and neuroprotective. Aggregation of neurotoxic proteins in NDDs, like amyloid-β (in case of AD)[202], tau (in case of AD)[202], α-synuclein (in case of PD), mSOD1 (in case of ALS), and TDP-43 (in case of ALS), initiate changes to induce both microglia and astrocytes to produce harmful pro-inflammatory pathological phenotypic biomarkers. The major pro-inflammatory biomarkers induced by both the glial cells implicated in the NDDs are interleukin (IL-1β), tumor necrosis factor (TNF-α), interleukin (IL-6), nitric oxide (NO)[200, 203]. Eventually, the production of these pro-inflammatory mediators leads to the progression of NDDs. Recent studies have shown that Bivalent (Ca) plays an indispensable role in maintaining CNS homeostasis and aberrant Ca+2 signaling in the CNS leads to NDDs. This is largely evidenced by the abnormal Ca+2 dysregulation in the microglial and astrocytic cells of CNS, thus initiating a cascade of neuroinflammatory progression of the disease, like in AD[204, 205], PD[206, 207] and ALS[208, 209].

Recent studies and perspectives have shown that nuclear factor κB (NFκB) in the case of AD[210], PD[211] and ALS[212] induces both microglia[213, 214] and astrocytes[215, 216] to produce several pro-inflammatory mediators implicated in various NDDs[198, 217]. Further, free radical-generating enzymes such as cyclooxy-genase-2 (COX-2), NADPH oxidase, inducible nitric oxide synthase (iNOS), and lipoxygenase are also implicated in NF-κβ activation in the case of AD[210]. Survival of motor neurons with the decrease in misfolded SOD1 protein is reported recently, with neuronal inhibition of NF-κB activity in SOD1G93A ALS mice model [218]. Together, enormous studies have portrayed the new modulatory role of NF-κB in instigating inflammation in the brain tissues [210, 219-223].

  1. The quality of the figures is overall poor and thus needs to be significantly improved.

Authors Comments: We have made all the efforts in making the schematic diagrams and figures to catch the attention of readers and the wide scientific community. We have used the most widely accepted tool Biorender, with a subscription to make all the figures, with high resolutions. We have also, appended all the figures while submitting the manuscript.

  1. It should be revised for grammar and mistyping overall the text. Some examples:

- Line 9: “alzheimer’s” should be replaced with “Alzheimer’s”.

- Line 9: “parkinson’s” should be replaced with “Parkinson’s”.

- Line 167: “(Na)” should be replaced with “(Na)

Authors Comments: We have revised the manuscript diligently, with your suggestions.

Reviewer 3 Report

I do appreciate the attempt by the authors to find common pathogenic mechanisms that govern a variety of different neurodegenerative disorders. However, those mechanisms described here are not novel or unique. Oxidative stress, protein homeostasis, autophagy, mitochondrial quality control, axonal transport,,, these all have already been very well implicated, and the authors do not seem to add novel or additional insights. The referenced papers are not from the most recent studies. From these, I do not exactly understand what the authors want to do here. I would advise to either focus on more restricted topics and/or to discuss most recent papers.

Author Response

Dear Reviewer,

My Sincere gratitude to you.

I want to thank and appreciate you, for reviewing our manuscript and making all the effort to provide valuable comments and suggestions for the improvement of the manuscript.

Below are the comments (highlighted in bold) by the reviewer and the revisions (highlighted in red in the manuscript and below) here and in the manuscript according to the recommendations of the reviewers.

  1. I do appreciate the attempt by the authors to find common pathogenic mechanisms that govern a variety of different neurodegenerative disorders. However, those mechanisms described here are not novel or unique. Oxidative stress, protein homeostasis, autophagy, mitochondrial quality control, and axonal transport, all have already been very well implicated, and the authors do not seem to add novel or additional insights. The referenced papers are not from the most recent studies.

Authors Comments: I want to thank you for your comments and suggestions to improve the manuscript. I would like to draw your attention to the below points regarding the uniqueness of the current manuscript, and we also did some improvements according to the suggestions and recommendations.

  • I accept your views that the pathological pathways represented in the manuscript are well-known. Here in this manuscript, we tried to interconnect the dots and did all the hard work to present the crucial cross-talks between various biochemical and molecular mechanisms across various NDDs, like ALS, AD, and PD. Further, we have cited highly relevant research with highly cited papers (high impact) in this review.

  • The figures were represented in a schematic manner to understand various biochemical pathways. We believe that without going in-depth into the literature, the readers could be able to connect the missing links and understand the overlapping pathways in various NDDs.

  • The review has given emphasis on plain language, which could catch the interest of the scientific community including biologist and medicinal chemist, who wants to target NDDs with better understanding.

Areas of Improvements

  • We have now added more than 30 relevant citations throughout the manuscript, making it a total of 223.
  • We have added an additional paragraph delineating the role of neuroinflammation in NDDs.
  • We have also, added our perspective and advocacy in the conclusion section.
  • We have done corrections in spelling and typos in the revised manuscript.

Further, we believe this manuscript will catch the attention of the scientific community because we have used high-resolution figures made from the Biorender tool.

Round 2

Reviewer 1 Report

The revised manuscript is improved and the discussion is more holistic. The authors are applauded for taking the comments of the reviewers seriously and committing to address them as fully as possible. 

Reviewer 2 Report

The manuscript has improved with the neuroinflammation paragraph. Although not exhaustive, it links well with the previous elements that, otherwise, would have been just self-standing.

Overall, the review appears scientifically sound and well written.

English is fundamentally ok, still needs some minor revision to optimize.

Author Response

Dear Reviewer,

My sincere gratitude to you.

Thank you for giving your positive thoughts and comments on the revised manuscript. We really appreciate your efforts to proof read the revised version.

We have extensively reviewed our manuscript for English corrections, highlighted in red in the entire manuscript.

Thank you and have a wonderful day ahead.

Reviewer 3 Report

While the scientific merit of this manuscript does not seem high, this may be acceptable.